# Detecting and Mitigating Hallucinations in Multilingual Summarisation

[1]Yifu Qiu    [1]Yftah Ziser    [2]Anna Korhonen    [1]Edoardo M. Ponti    [1]Shay B. Cohen
[1]Institute for Language, Cognition and Computation, University of Edinburgh
[2]Language Technology Lab, University of Cambridge
{yifu.qiu,yftah.ziser,eponti,scohen}@ed.ac.uk

## Abstract

Hallucinations pose a significant challenge to the reliability of neural models for abstractive summarisation. While automatically generated summaries may be fluent, they often lack faithfulness to the original document. This issue becomes even more pronounced in low-resource languages, where summarisation requires cross-lingual transfer. With the existing faithful metrics focusing on English, even *measuring* the extent of this phenomenon in cross-lingual settings is hard. To address this, we first develop a novel metric, mFACT, evaluating the faithfulness of non-English summaries, leveraging translation-based transfer from multiple English faithfulness metrics. Through extensive experiments in multiple languages, we demonstrate that mFACT is best suited to detect hallucinations compared to alternative metrics. With mFACT, we assess a broad range of multilingual large language models, and find that they all tend to hallucinate often in languages different from English. We then propose a simple but effective method to *reduce* hallucinations in cross-lingual transfer, which weighs the loss of each training example by its faithfulness score. This method drastically increases both performance and faithfulness according to both automatic and human evaluation when compared to strong baselines for cross-lingual transfer such as MAD-X. Our code and dataset are available at https://github.com/yfqiu-nlp/mfact-summ.

## 1 Introduction

Recent neural abstractive summarisation models (Lewis et al., 2020; Liu and Liu, 2021; Liu et al., 2022; Fonseca et al., 2022; Ravaut et al., 2022) have shown promise in terms of ROUGE scores (Lin and Och, 2004). However, a well-known problem with these models is *hallucination* (Maynez et al., 2020; Kryscinski et al., 2020; Laban et al., 2022; Cao et al., 2022a)—generating summaries that cannot be supported by ground-truth knowl-

edge (e.g., news, documents, meeting notes). Anecdotally, it was shown that the percentage of generated summaries for CNN/DailyMail (Hermann et al., 2015; See et al., 2017) and XSum (Narayan et al., 2018) containing hallucinations amounts to up to 74.8% and 96.9% (Pagnoni et al., 2021), respectively. Hallucinations hinder the reliability of abstractive summarisation systems by potentially misleading users with the misinformation they produce.

In addition, current summarisation models, open-source or proprietary, struggle in low-resource settings (Parida and Motlicek, 2019; Hasan et al., 2021; Bai et al., 2021; Urlana et al., 2023), when the target language is under-represented (e.g., Vietnamese and Urdu). Fortunately, cross-lingual transfer methods (Pfeiffer et al., 2020b; Xue et al., 2021; Hu et al., 2020) leverage task-specific knowledge learned from a resource-rich source language to summarise documents in many resource-poor target languages, in a zero-shot fashion or only with few annotated examples. Nevertheless, it remains unclear to what extent cross-lingual summarisation suffers from the problem of hallucination, compared to monolingual systems where English is the only language.

The main challenge in addressing this question is that most faithfulness evaluation metrics are available only for English and do not support low-resource languages. Hence, our first contribution (Section 2) is a model-based metric (mFACT) that measures the factual consistency of multilingual conditional generation, obtained from four diverse English faithfulness metrics (Goyal and Durrett, 2021; Fabbri et al., 2022; Cao et al., 2022a) via 'translate train' knowledge transfer (Artetxe et al., 2020). As illustrated in Figure 1, we use existing faithfulness metrics to label the English document–summary pairs as positive (i.e., faithful) or negative (i.e., hallucinated) and translate them into each target language. We then train a classifier in each

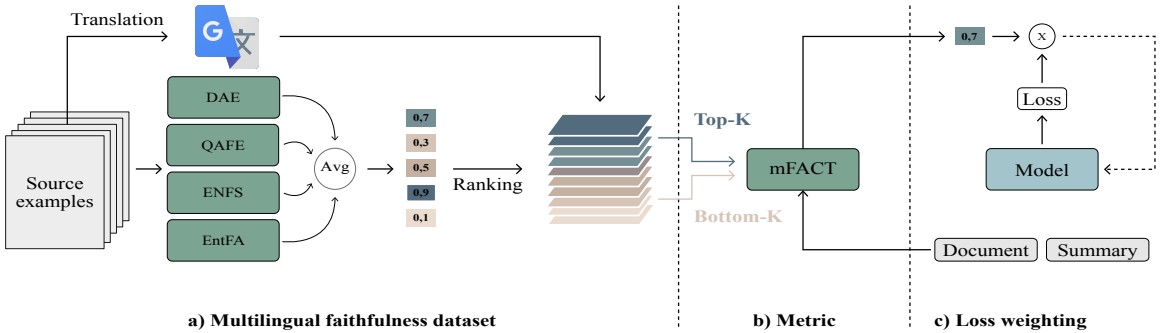

**a) Multilingual faithfulness dataset**  **b) Metric**  **c) Loss weighting**

Figure 1: Pipeline of mFACT for transferring English faithfulness metrics to target languages via machine translation. We average the score of four English metrics to rank the training samples in XSum. We then translate the most faithful and hallucinated samples into each target language and train a classifier to distinguish them.

target language to predict the faithfulness scores for the translated document–summary pairs. We verify the reliability of mFACT on the translated test set and, most importantly, with human evaluation. These confirm the effectiveness of mFACT in capturing hallucinations in target languages.

Equipped with this new metric, we conduct extensive cross-lingual transfer experiments on XL-Sum (Hasan et al., 2021) for abstractive summarisation in six typologically diverse languages: Chinese, Spanish, French, Hindi, Turkish and Vietnamese. We find that state-of-the-art cross-lingual transfer methods increase summarisation performance in the target languages, *but* also introduce more hallucinations compared to English monolingual models in comparable experimental settings, thus further exacerbating this tendency (Section 6).

We also employ the mFACT metric to assess the faithfulness of some recently released multilingual large language models (LLMs), including Phoenix, BLOOMZ, and Vicuna (Chen et al., 2023; Muennighoff et al., 2022; Chiang et al., 2023; Le Scao et al., 2022). We show that LLMs that use multilingual data for pre-training or conversational fine-tuning fail to ensure faithfulness in summarisation in various languages, producing more hallucinations in low-resource ones.

To overcome this limitation and promote faithful summarisation in multiple languages, we adapt a series of existing methods for reducing hallucinations originally devised for monolingual summarisation (Section 3.2). In addition, we introduce a novel, simple but effective method (Section 3.3): we weigh the loss for each training example according to their faithful scores. We evaluate our loss-weighting method with automated metrics and

human judgements. We observe significant gains in both summarisation performance and faithfulness over a series of strong baselines (Section 8). In a nutshell, our main contributions are the following:

- We propose mFACT, a multilingual faithful metric developed from four English faithfulness metrics. This enables detecting hallucinated summaries in languages other than English.
- To the best of our knowledge, we are the first to study hallucination in a cross-lingual transfer setting. We show that state-of-the-art methods like MAD-X (Pfeiffer et al., 2020b) can improve the performance for low-resource summarisation, but also amplify hallucinations.
- We apply mFACT to study the faithfulness in summarisation of the recent multilingual Large Language Models. We observe that despite their scale, these models are still struggling to reduce hallucinations for languages other than English.
- We propose a novel method to enhance faithfulness and performance in cross-lingual transfer for summarisation, which consists of weighting training samples' loss based on their faithfulness score. Both automatic and human evaluations validate the superiority of our method over existing baselines.

## 2 mFACT: A Multilingual Metric for Faithfulness

The lack of faithful metrics in languages other than English greatly limits the evaluation (and hence, the prevention) of hallucinations in cross-lingual transfer for low-resource languages. In this section, we fill this gap by introducing mFACT. This is constructed by transferring multiple English faithful

metrics into *any* target language, given the availability of a machine translation model.

## 2.1 Translation-based Transfer for Faithfulness Metrics

One straightforward way to implement a faithful metric in any target language is by implementing it from scratch following the design of monolingual English metrics. However, these often rely on data annotated with auxiliary language-specific tools. For instance, Dependency Arc Entailment (DAE; Goyal and Durrett 2021) requires an external dependency parser to label fine-grained hallucinated segments. This is impractical due to the lack of annotated data and auxiliary tools in most languages. Another strategy relies on "translate test" knowledge transfer (Artetxe et al., 2020), where test documents and their corresponding generated summaries are translated from the target language to English. Then, English metrics can measure faithfulness; however, this introduces noise from translation and is costly at inference time, which makes this unsuitable for model development. For instance, model selection is commonly based on early stopping according to validation faithful scores (Choubey et al., 2021; Aharoni et al., 2022), which necessitates translating all generated summaries at each validation step.

Our solution instead is to formulate faithfulness evaluation as a binary classification problem, i.e., to predict whether a given document–summary pair is faithful or hallucinated. In other terms, our proposed approach aims to distil knowledge from multiple teacher models, i.e., existing English model-based metrics, into a target-language classifier as a student model. Specifically, we use multiple English faithful metrics to assign the pseudo labels of "faithful" or "hallucinated" for English document–summary pairs, then translate them to create a faithfulness binary classification dataset in target languages. We then train the target-language classifier on the resulting silver dataset. Formally, we aim to obtain a faithfulness scoring model $g(\cdot)$ in target language $tgt$ that predicts the faithfulness for a given document-summary pair $(\mathbf{x}, \mathbf{y})$. Hence $g^{(tgt)}(\mathbf{x}^{(tgt)}, \mathbf{y}^{(tgt)}) \triangleq p(z = 1 \mid \mathbf{x}^{(tgt)}, \mathbf{y}^{(tgt)})$ where $z = 1$ and $z = 0$ represent whether the pair is faithful or hallucinated, respectively.

The pipeline for creating mFACT is presented in Figure 1. We start with four diverse English faithfulness metrics[1], and use them to score the training samples from the English XSum summarisation dataset (Narayan et al., 2018). Following Maynez et al. (2020), we select the metrics based on two categories of hallucinations generated by the model: 1) *intrinsic hallucinations* where the summary distorts the information present in the document; 2) *extrinsic hallucinations* where the model adds information that cannot be directly supported by the document. We select two model-based metrics capturing intrinsic hallucinations:

- **DAE** (Goyal and Durrett, 2021) which consists in an entailment classifier trained with annotation at a fine-grained dependency level;
- **QAFactEval** (Fabbri et al., 2022) which focuses on generating questions whose answer is a span of the summary, and attempts to answer them based on the document alone;

and two metrics for extrinsic hallucinations:

- **ENFS%** is a simple rule-based measurement presented by Cao et al. (2022a), which counts the proportion of entities which appear in a summary but not in its corresponding document.
- **EntFA** (Cao et al., 2022a) which estimates the posterior and prior probabilities of generated entities with language models conditioned (or not conditioned, respectively) on the source document. Using these probabilities as features, a KNN classifier detects token-level hallucination.

We chose XSum as the source English dataset because 1) all our selected faithfulness metrics are trained on XSum, which allows us to maximise the reliability of these metrics. 2) XSum has been shown to include abundant and diverse hallucinations (Maynez et al., 2020; Pagnoni et al., 2021), which allows our metrics to capture as many types of hallucinations as possible.

We then normalise the scores from the above-mentioned four metrics between $[0, 1]$ and average them for each training sample. We rank the samples from the most faithful to the most hallucinated according to the resulting faithfulness scores. The $k$ top-ranked and $k$ bottom-ranked document–summary pairs are then considered positive and negative examples, respectively. We translate these into a series of target languages with the Google Translation API[2] and create our silver faithfulness dataset splitting its examples with a proportion of $95/2.5/2.5$ as the training/validation/testing sets.

---

[1] Our simple sanity check in Appendix A.3 shows these model-based hallucination metrics to be reliable.

[2] https://cloud.google.com/translate

Finally, a multilingual BERT-based classifier is fine-tuned on our dataset. We follow the sentence-pair classification setting from (Devlin et al., 2019) to concatenate the document–summary pairs as the input. A classifier receives the last-layer representation for the [CLS] special token and returns a score between 0 (hallucinated) and 1 (faithful).

# 3 Reducing Hallucination in Cross-lingual Transfer

We first provide some background on cross-lingual transfer. Then, we show how to adapt several methods promoting faithfulness in monolingual summarisation to cross-lingual transfer settings. Finally, we describe a new approach based on loss weighting.

## 3.1 Cross-lingual Transfer with MAD-X

We adopt the Multiple ADapters framework (MAD-X; Pfeiffer et al. 2020b), which constitutes a state-of-the-art method for cross-lingual transfer. MAD-X learns independent language and task adapters (i.e., parameter-efficient model fine-tunings), and then combines them. Specifically, to transfer the ability to summarise documents from a source language to a target language, we follow these steps: 1) We train two separate language adapters on the Wikipedia corpora for both the source and target languages. 2) We stack the (frozen) source language adapter with a randomly initialised task adapter and train the latter with annotated data in the source language. 3) We stack the trained task adapter with the target language adapter and then perform zero-shot inference in the target language.

## 3.2 Expert and Anti-Expert Approaches

The majority of strategies to reduce hallucinations in monolingual settings rely on creating experts or anti-experts that steer the model towards positive behaviour or away from negative behaviour. As a by-product of the pipeline to create our metric, mFACT (Section 2), we obtained two separate subsets of faithful and hallucinated samples in both source and target languages. These subsets can serve as training data for experts/anti-experts in multiple languages, thus making them suitable for cross-lingual transfer. We explore three methods in this family. In all in stances, we first train a base adapter with the source summarisation dataset. Then, we further tune it with the faithful (hallucinated) subset to obtain an *expert (anti-expert)*

adapter.

**Task Vector Negation (TVN; Ilharco et al. 2022).** Task vector negation mitigates hallucinated generation by subtracting the *task vector* of the anti-expert model from the fine-tuned model. Formally, given a fine-tuned model with parameter $\boldsymbol{\theta}_0$ and an anti-expert model $\boldsymbol{\theta}^-$, the interpolated model parameters $\boldsymbol{\theta}^\star$ are obtained as

$$\boldsymbol{\theta}^\star = \boldsymbol{\theta}_0 - \lambda(\boldsymbol{\theta}^- - \boldsymbol{\theta}_0), \qquad (1)$$

where $\lambda$ is the importance hyperparameter that controls the degree of fusion between the fine-tuned model and the anti-expert.

**Contrastive Parameter Ensembling (CAPE; Choubey et al. 2021).** To compensate for the potential loss of summarisation ability by only subtracting the anti-expert task vector from the base model, CAPE proposes to also add the expert parameters. Formally, the interpolated model parameters $\boldsymbol{\theta}^\star$ are obtained as:

$$\boldsymbol{\theta}^\star = \boldsymbol{\theta}_0 + \lambda(\boldsymbol{\theta}^+ - \boldsymbol{\theta}^-), \qquad (2)$$

where $\lambda$ again is the importance hyperparameter.

**DExpert Decoding (Liu et al., 2021).** Contrary to Task Vector Negation and CAPE, which directly manipulate the model parameters, DExpert uses expert and anti-expert models to modify the predicted logits at each decoding step. Given the base model $f_{\boldsymbol{\theta}}$ and a pair of expert $f_{\boldsymbol{\theta}^+}$ and anti-expert $f_{\boldsymbol{\theta}^-}$ models, the scores for the next token at each decoding step $t$ are:

$$p(\mathbf{y}_t|\mathbf{x}, \mathbf{y}_{<t}) = \text{softmax}(\mathbf{z}_t + \lambda(\mathbf{z}_t^+ - \mathbf{z}_t^-)), \quad (3)$$

where $\mathbf{z}_t, \mathbf{z}_t^+, \mathbf{z}_t^-$ are the outputs from $f_{\boldsymbol{\theta}}, f_{\boldsymbol{\theta}^+}, f_{\boldsymbol{\theta}^-}$ at time step $t$, respectively. Again, an importance hyper-parameter $\lambda$ controls the degree of fusion during decoding.

## 3.3 Weighted Loss Approach

We also introduce a simple but effective approach to reduce hallucination during cross-lingual transfer. Previous works have shown that controlling the quality of the training samples can improve the model's faithfulness (Kang and Hashimoto, 2020; Aharoni et al., 2022). However, simply filtering out hallucinated training data may sacrifice the summarisation performance (Dziri et al., 2022).

We thus propose a "soft" data filtering approach where we weigh the training loss according to each sample's faithfulness score. More formally, we rely

| Model | Acc. | | Prec. | | Recall | | F1 | |
|---|---|---|---|---|---|---|---|---|
| | $\mu$ | $\sigma$ | $\mu$ | $\sigma$ | $\mu$ | $\sigma$ | $\mu$ | $\sigma$ |
| XNLI | 52.9 | 0.7 | 71.6 | 3.3 | 8.4 | 1.5 | 15.0 | 2.5 |
| X.-mF | 95.0 | 2.3 | 95.7 | 2.4 | 94.3 | 2.2 | 94.9 | 2.3 |
| mF-T | 64.2 | 6.2 | **98.9** | 0.9 | 28.3 | 12.9 | 42.5 | 16.9 |
| mF | **95.3** | 1.7 | 95.4 | 1.8 | **95.1** | 2.0 | **95.2** | 1.8 |

Table 1: Mean values ($\mu$) and standard deviations ($\sigma$) of the test performance of four faithfulness classifiers over six target languages. X.-mF, mF-T, mF stand for XNLI-mFACT, mFACT-Transfer and mFACT, respectively. The detailed results for each language are given in Appendix A.5.

on a faithfulness metric for the source language, which outputs a score $z^{(i)}$ for the $i^{th}$ document–summary pair's faithfulness. Then the update rule of training parameters for each batch becomes

$$\boldsymbol{\theta}^* = \boldsymbol{\theta} - \alpha \left( \frac{1}{m} \sum_{i=1}^{m} z^{(i)} \nabla_{\boldsymbol{\theta}} J(\mathbf{x}^{(i)}, \mathbf{y}^{(i)}; \boldsymbol{\theta}) \right), \tag{4}$$

where $\boldsymbol{\theta}$ is the vector of trainable model parameters, $\alpha$ is the learning rate, $m$ is the batch size, $J(\cdot; \boldsymbol{\theta})$ is the loss function for a single training example $(\mathbf{x}^{(i)}, \mathbf{y}^{(i)})$, and $\nabla_{\boldsymbol{\theta}} J(\cdot)$ is the gradient of the loss function wrt. the model parameters.

## 4 Experimental Setup

**Evaluation Metrics.** We use ROUGE-1/2/L scores (Lin and Och, 2004) to evaluate the task of abstractive summarisation. We use the four metrics mentioned in Section 2 to evaluate the faithfulness of English summaries and our mFACT metric for summaries in other languages.

**Dataset.** We conduct our experiments on XL-Sum, which is a large-scale multilingual summarisation dataset (Hasan et al., 2021). XL-Sum provides a large collection of annotated document–summary pairs in 45 languages in addition to English. We test our approach on six target languages: Chinese, Spanish, French, Hindi, Turkish and Vietnamese. Table 7 shows the dataset statistics.

## 5 Faithfulness Classification Experiments

### 5.1 Classification Results

Firstly, we verify the reliability of mFACT by using our translated test sets in multiple languages to benchmark mFACT and several baselines for faithfulness classification.

**Baselines.** Previous works (Maynez et al., 2020; Kryscinski et al., 2020) showed that models train

| Models | R-1↑ | DAE↑ | QAFE↑ | ENFS↓ | EntFA↑ |
|---|---|---|---|---|---|
| MAD-X | 23.62 | 80.14 | 52.12 | 23.04 | 93.09 |
| XNLI | 23.42 | 84.92 | 53.42 | 21.56 | 94.23 |
| XNLI-mFACT | 23.64 | **86.48** | 52.26 | 21.50 | 94.33 |
| mFACT-TF | **23.97** | 84.25 | 53.14 | 21.16 | 94.51 |
| mFACT | 23.24 | 85.29 | **54.30** | **20.68** | **94.63** |

Table 2: Results for (inverse) cross-lingual transfer *from* other languages to English. We report the average values for performance (R-1) and faithfulness (DAE, QAFE, ENFS%, EntFA) metrics. The methods include the MAD-X baseline and our proposed loss weighting with four weighting metrics, including an XNLI-trained classifier and mFACT. Numbers are averages of 3 different random seeds for each of the 6 languages.

for natural language inference (NLI), a related task for which more annotated data is readily available, can be used also for assessing faithfulness for English summarisation. We thus include a baseline, namely XNLI, which consists in fine-tuning multilingual BERT with the corresponding language split in the XNLI dataset (Conneau et al., 2018). As an alternative, we further fine-tune the XNLI baseline with our translated data (XNLI-mFACT), thus verifying whether combining the supervision signal from both sources boosts the performance. Finally, we incorporate an ablation study for using zero-shot multilingual transfer instead of "translate train" (Artetxe et al., 2020). In mFACT-Transfer, we train a multilingual encoder on our English faithfulness classification dataset *without translating it*, then deploy it directly on examples in other languages.

**Results and Discussion.** We report the classification performance in Table 1. We find that NLI classifiers do not achieve a level of performance on par with classifiers trained on our faithfulness classification dataset. This demonstrates that evaluating faithfulness is indeed distinct from NLI, which is consistent with previous findings in assessing faithfulness in English (Kryscinski et al., 2020; Maynez et al., 2020). Comparing mFACT and mFACT-Transfer, we also observe the positive effects of translation-based transfer, which achieves a much higher recall rate than zero-shot cross-lingual transfer. Hence, mFACT is more likely to identify faithful document–summary pairs as such.

### 5.2 External Evaluation by Inverse Transfer

Finally, we conduct an evaluation based on *inverse* cross-lingual transfer (i.e., from other languages to English) as a downstream task with our newly intro-

duced approach (Section 3.3). This setting allows us to compare the impact of using different multilingual faithfulness metrics, among those listed in Section 5.1, to weigh the training samples in target languages. The logic behind this experiment is that if the scorer captures the model's faithfulness in target languages, the English summaries generated by the corresponding model should be more faithful according to the four English metrics from Section 2.1.

The results are shown in Table 2. Unsurprisingly, we observe that in general weighting the training samples in target languages with faithfulness metrics can achieve considerable improvements over the MAD-X baseline on English faithfulness scores. This suggests that these metrics are well aligned with the actual faithfulness of generated summaries. Specifically, comparing mFACT-Ours and mFACT-Transfer methods with XNLI and XNLI+mFACT, we find that our constructed dataset is much more effective in improving faithfulness than NLI signal, which again verifies our previous assumption that faithfulness classification and NLI are only vaguely related. Finally mFACT-Transfer performs worse than mFACT in ROUGE, which can be caused by the much lower recall rate of mFACT-Transfer in faithfulness classification (see Table 1).

## 6 Cross-lingual Transfer Introduces Additional Hallucinations

The second analysis of this paper aims to corroborate our observation that cross-lingual transfer can introduce *additional* hallucinations over monolingual fine-tuning, though it improves the task performance for summarisation in the target language.

**Transfer Setup.** We compare two data scenarios and two styles of fine-tuning. To begin, we investigate the impact of initial training on source data, followed by applying few-shot learning techniques on target data (cross-lingual transfer) instead of direct application. We attribute the difference in faithfulness scores to the additional hallucinations introduced by the training phase in the source language. Taking Chinese as an example of few-shot cross-lingual transfer, we train the summarisation model first on XL-Sum (Chinese) and then with 1K randomly sampled XSum (English) examples. Secondly, we compare fine-tuning the full model, where all parameters are updated, with parameter-efficient fine-tuning, where only the adapters are updated. This allows us to study the effect of dif-

| Metrics | | MAD-X | | Full Model | |
|---|---|---|---|---|---|
| | | MFT | CLTF | MFT | CLTF |
| Perform. | R-1 | 30.25 | **30.96** | 23.96 | **32.05** |
| | R-2 | 8.57 | **9.11** | 5.9 | **9.69** |
| | R-L | 22.48 | **23.14** | 17.96 | **23.85** |
| Faithful. | DAE (↑) | **68.17** | 66.69 | **84.33** | 53.24 |
| | QAFE (↑) | **34.44** | 33.87 | **63.52** | 30.98 |
| | ENFS% (↓) | **33.29** | 35.07 | **16.82** | 41.45 |
| | EntFA (↑) | 87.58 | **87.87** | **95.14** | 82.96 |

Table 3: Performance and faithfulness scores for few-shot cross-lingual transfer (CLTF) and monolingual fine-tuning (MFT) on abstractive summarisation. CLTF generally improves the model's performance but decreases its faithfulness. ↑ and ↓ indicate higher or lower values are better, respectively.

ferent transfer methods on faithfulness.

**Results and Discussion** In Figure 3, we observe that cross-lingual transfer improves ROUGE scores for both full-model fine-tuning and MAD-X, outperforming monolingual fine-tuning. This underscores its effectiveness in transferring task-specific knowledge from source to target languages in low-resource scenarios. However, it's important to note that leveraging source language data can also increase hallucination in both cases.

## 7 Hallucinations in Multilingual Large Language Models

We also assess the summarisation performance of recent multilingual large language models (LLMs) on XL-Sum in Table 5. We carefully select three representative multilingual LLMs for investigation,

- **BLOOMZ-P3-7.1B** (Muennighoff et al., 2022) represents the instruction-tuned model with English P3 dataset, which derives from the multilingual BLOOM (Le Scao et al., 2022). We decide not test BLOOMZ-xP3 trained with machine-translated instructions from English P3 because we consider this experiment as an assessment to the cross-lingual transfer capabilities from the multilingual model.
- **Vicuna-7B** (Chiang et al., 2023) harnesses 70K multilingual conversation-style interactions to fine-tune LLaMA. Vicuna originates from the monolingual LLaMA, and the inclusion of Vicuna aims to test the cross-lingual transfer ability arising from multilingual conversational tuning.
- **Phoenix-7B** (Chen et al., 2023) is the current state-of-the-art, which continues to train

| L. | Method | R-1 | R-2 | R-L | mF | mF-T | bi% | tr% |
|---|---|---|---|---|---|---|---|---|
| Chinese | MAD-X | 29.59 | 14.86 | 20.61 | 39.62 | 35.08 | 21.62 | 34.37 |
| | CAPE | 29.64 | 14.80 | 20.58 | 38.83 | 34.01 | 20.11 | 32.32 |
| | TVN | 29.68 | 14.75 | 20.32 | 38.53 | 32.61 | 17.67 | 28.76 |
| | Dexpert | 29.59 | 14.86 | 20.61 | 39.63 | 35.08 | 21.62 | 34.37 |
| | Ours | **31.24** | **16.13** | **22.06** | **43.16** | **37.85** | **30.16** | **47.02** |
| Spanish | MAD-X | 23.36 | 5.13 | 16.34 | 21.87 | 29.36 | 21.98 | 34.20 |
| | CAPE | 23.24 | 5.01 | 16.24 | 21.65 | 29.40 | 19.98 | 31.29 |
| | TVN | 23.53 | 5.06 | 16.48 | 23.82 | 30.54 | 18.02 | 28.60 |
| | Dexpert | 23.36 | 5.13 | 16.34 | 21.88 | 29.36 | 21.98 | 34.20 |
| | Ours | **24.30** | **6.10** | **17.41** | 23.83 | 33.31 | **34.54** | **51.69** |
| Hindi | MAD-X | 25.51 | 7.78 | 19.07 | 28.41 | 19.32 | 25.57 | 39.02 |
| | CAPE | **25.80** | **7.85** | **19.20** | 29.11 | 19.53 | 23.77 | 36.58 |
| | TVN | 25.28 | 7.73 | 19.15 | **32.76** | **24.61** | 21.57 | 33.28 |
| | Dexpert | 25.51 | 7.78 | 19.07 | 28.40 | 19.32 | 25.57 | 39.02 |
| | Ours | 24.47 | 7.46 | 18.48 | 28.48 | 19.52 | **34.86** | **50.99** |
| Turkish | MAD-X | **17.22** | **6.33** | **14.59** | 33.24 | 25.53 | 38.72 | 54.12 |
| | CAPE | 17.12 | 6.23 | 14.55 | **35.04** | **26.69** | 36.47 | 51.47 |
| | TVN | 16.95 | 6.28 | 14.49 | 34.56 | 25.97 | 34.05 | 48.91 |
| | Dexpert | **17.22** | **6.33** | **14.59** | 33.22 | 25.53 | 38.72 | 54.12 |
| | Ours | 17.16 | 6.28 | 14.46 | 34.91 | 25.83 | **45.34** | **61.50** |
| Vietnamese | MAD-X | 27.23 | 12.57 | 20.32 | 36.64 | 37.75 | 27.40 | 42.67 |
| | CAPE | 27.01 | 12.45 | 20.15 | 36.71 | 37.79 | 25.89 | 40.64 |
| | TVN | 26.73 | 12.36 | 20.07 | **38.41** | **39.34** | 25.86 | 40.48 |
| | Dexpert | 27.23 | 12.57 | 20.32 | 36.61 | 37.75 | 27.40 | 42.67 |
| | Ours | **27.76** | **12.86** | **20.83** | 38.27 | 38.02 | **30.23** | **46.27** |
| French | MAD-X | 26.02 | 7.97 | 19.02 | 38.71 | **42.66** | 18.88 | 30.29 |
| | CAPE | 25.75 | 7.93 | 18.80 | 37.54 | 40.91 | 18.00 | 28.88 |
| | TVN | 25.54 | 7.86 | 18.71 | 38.18 | 41.74 | 17.37 | 27.68 |
| | Dexpert | 26.02 | 7.97 | 19.02 | **38.74** | **42.66** | 18.88 | 30.29 |
| | Ours | **27.70** | **9.09** | **20.27** | 36.83 | 39.75 | **33.81** | **50.57** |
| Average | MAD-X | 24.82 | 9.11 | 18.33 | 33.08 | 31.62 | 25.69 | 39.11 |
| | CAPE | 24.76 | 9.05 | 18.25 | 33.15 | 31.39 | 24.04 | 36.86 |
| | TVN | 24.62 | 9.01 | 18.20 | **34.38** | **32.47** | 22.42 | 34.62 |
| | Dexpert | 24.82 | 9.11 | 18.33 | 33.08 | 31.62 | 25.69 | 39.11 |
| | Ours | **25.44** | **9.65** | **18.92** | 34.25 | 32.38 | **34.82** | **51.34** |

Table 4: Automatic evaluation for **zero-shot** cross-lingual transfer performance from English to other languages when selecting the checkpoint with the best validation mFACT. Numbers represent the average of three runs with different random seeds. mF stands for mFACT and mF-T stands for mFACT-Transfer. bi% and tr% stand for the percentages of novel bigrams and trigrams.

| | Lang. | P. (%) | R-1 | R-L | mF. |
|---|---|---|---|---|---|
| Phoenix | English | **58.6** | **29.98** | **21.11** | **47.22** |
| | French | 2.1 | 29.33 | 18.80 | 24.03 |
| | Spanish | 3.0 | 15.92 | 10.65 | 18.69 |
| | Hindi | 1.1 | 2.70 | 2.64 | 16.53 |
| Vicuna | English | / | **31.74** | **22.51** | **57.3** |
| | French | / | 23.82 | 15.99 | 23.02 |
| | Spanish | / | 12.92 | 7.42 | 23.68 |
| | Hindi | / | 1.30 | 1.29 | 13.53 |
| BLOOMZ | English | 30.0 | 17.07 | 10.89 | **53.70** |
| | French | 12.9 | **23.17** | **13.97** | 26.34 |
| | Spanish | 10.8 | 8.54 | 4.64 | 27.90 |
| | Hindi | 1.3 | 8.22 | 7.80 | 12.52 |

Table 5: Assessing the multilingual summarisation performance for Vicuna-7B, Phoenix-7B, and BLOOMZ-7.1B on four languages (**Lang.**) using ROUGE-1/L (**R-1/L**) and mFACT (**mF.**) metrics. We also report the percentage (**P. (%)**) of samples in each language an LLM was exposed to during their multilingual training.

multilingual training.

Table 5 demonstrates that current LLMs display notable faithfulness limitations in cross-lingual transfer contexts for languages beyond English, including well-resourced languages like French and Spanish. Furthermore, a noticeable trend emerges: LLM faithfulness across languages tends to correlate highly to the number of samples from target languages observed during their training. These observations align with recent findings (Lai et al., 2023; Laskar et al., 2023) which highlight the challenges in maintaining faithfulness while generating content in low-resource languages.

## 8 Reducing Hallucinations

In this section, we test different methods for cross-lingual transfer of summarisation to multiple languages and for promoting faithfulness. We compare our new method of loss weighting based on mFACT with MAD-X, as well as with a series of approaches for reducing hallucinations (Section 4). We evaluate these methods with automated metrics for performance, faithfulness, and abstractiveness (i.e., the ability to rephrase the document instead of copy-pasting spans of text). We also conduct human evaluations to corroborate these results.

**Automatic Evaluation.** We report ROUGE scores for performance, faithfulness (mFACT), and abstractiveness (novel bigrams and trigrams in the

BLOOMZ with an additional 267K and 189K instances of multilingual instructions and conversation rounds.

We select three languages, aside from English, which are present in the pre-training data for BLOOMZ and the conversational tuning data for Vicuna and Phoenix. We also report the percentage of examples in each of these languages that these models have been exposed to during their

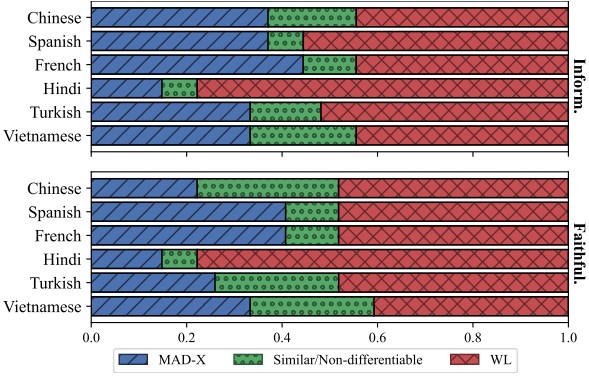

Figure 2: Human preferences for summaries in terms of Inform[ativeness] and Faithful[ness]. Annotators could choose MAD-X, weighted loss (WL, ours), or non-differentiable (if summaries are too similar).

|  | **Pearson** | | **Spearman** | |
| --- | --- | --- | --- | --- |
|  | $\rho$ | $p$ **value** | $\rho$ | $p$ **value** |
| XNLI | 0.22 | 0.10 | 0.25 | 0.07 |
| XNLI-mF | 0.25 | 0.07 | 0.28* | 0.04 |
| mF-T | 0.44* | 0.00 | **0.36*** | 0.01 |
| mF | **0.45*** | 0.00 | 0.34* | 0.01 |

Table 6: Correlation between several faithfulness metrics and human preferences. mF and mF-T stand for mFACT and mFACT-Transfer, respectively. We calculate both Pearson and Spearman statistics on document–summary pairs from all six languages to ensure that the sample size is significant.

summary) for the test set of each target language in Table 4. We first observe that the expert/anti-expert methods adapted from monolingual summarisation are partly effective for improving ROUGEs and mFACT score in cross-lingual transfer over MAD-X; however, no clear winner emerges among them, as their gains are marginal or inconsistent. For example, TVN produces the most faithful summaries for Hindi and Vietnamese, CAPE for Turkish, and DExpert for French. All three models, however, display a similar trend of sacrificing ROUGE scores to improve faithfulness. Instead, as Table 4 demonstrates, our proposed weighted-loss approach (WL) improves the performance across the board while achieving a comparable mFACT score with the most faithful expert models. In particular, WL achieves the best faithfulness in Chinese and Spanish and the best ROUGE scores for all languages except Hindi. These results suggest that our weighted-loss method strikes the best balance between summarisation abilities and faithfulness.

**Abstractiveness.** We also measure the levels of ab-

stractiveness of different methods, which is known to be inversely correlated with faithfulness (Ladhak et al., 2022; Daheim et al., 2023). In fact, reducing hallucinations has the side effect of encouraging the model to copy-paste spans of the document (i.e., acquiring an extractive behaviour). Following Cao et al. (2022a) and See et al. (2017), we use the percentage of novel $n$-grams in summaries compared with the document as a measure of abstractiveness.

Figure 3 illustrates the distributions of abstractiveness and faithfulness for all models in six XL-Sum datasets. Both positive and negative predictions of mFACT scatter with different levels of abstractiveness. We also observe that summaries generated by the weighted loss method generally have a higher level of abstractiveness when they are similarly faithful compared with other baselines. Table 4 shows most expert/anti-expert models sacrifice abstractiveness to improve faithfulness score. In contrast, the weighted loss approach produces more novel $n$-grams. These findings show that our method does not improve faithfulness by simply favouring extractive summaries.

**Human Evaluation.** Finally, we recruited human annotators from the Prolific platform[3] for a blind comparison between MAD-X and our weighted-loss model. We randomly sampled nine documents for each language and paired them with the summaries generated by the two models. We asked the human participants to evaluate the summaries via A/B testing in two aspects,

> **Informativeness**: *An informative summary should cover as much information from the document as possible, while it should convey the main idea of the document.*
> **Faithfulness**: *A faithful summary should only contain information already present in the document* [4] *and should not contain information contradicting the document.*

Participants will first read the document, then select the better summary (or both, if they are similar) in terms of informativeness and faithfulness (see Appendix A.5). We require participants to be native speakers of the language they evaluate and have obtained at least a bachelor's degree. Each document and its paired summaries are evaluated by 3 participants. These settings allow us to achieve a fair inter-rater agreement of 0.28 in terms of Fleiss' $\kappa$ (Landis and Koch, 1977).

The results in Figure 2 indicate that human

---

[3] https://app.prolific.co

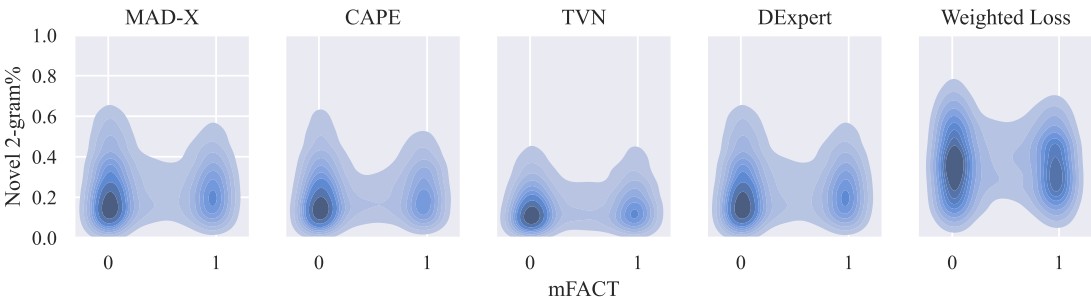

Figure 3: Distributions for Novel 2-gram% and mFACT scores for all five hallucination reduction methods in cross-lingual transfer for the datasets of 6 languages in XL-Sum.

evaluators prefer the summaries generated by our weighted loss method rather than MAD-X, demonstrating that our weighted loss approach improves faithfulness and informativeness for all six languages.

Finally, we study the correlation between the human preferences from Figure 2 and various faithfulness metrics presented in Section 5.1. From Table 6, it emerges that mFACT achieves the strongest correlation with human judgements (0.45 Pearson $\rho$ and 0.34 Spearman $\rho$), which is statistically significant. In comparison with XNLI and XNLI-mF, we reconfirm that metrics designed for faithfulness classification, rather than natural language inference, more effectively align with human preferences.

## 9 Related Work

While faithfulness in summarisation is a highly researched topic, previous works focused mostly on the English language (Pagnoni et al., 2021; Maynez et al., 2020; Fabbri et al., 2021). To evaluate faithfulness, state-of-the-art methods fall into three categories. Firstly, validating faithfulness can be cast as a classification problem (Kryscinski et al., 2020; Goyal and Durrett, 2021; Laban et al., 2022). Secondly, faithfulness can be interpreted as answerability, and assessed with existing question answering models (Fabbri et al., 2022; Scialom et al., 2021). Finally, language models may be adopted to identify extrinsic hallucinations (Filippova, 2020; Cao et al., 2022a). To improve faithfulness in summarisation models, an approach related to ours changes the training dynamics, e.g. by filtering out hallucinated data (Cao et al., 2022b; Kang and Hashimoto, 2020; Goyal et al., 2022). The expert/anti-expert approach aims to learn experts and anti-experts that alter the behaviour of a base generative model (Liu et al., 2021; Choubey et al., 2021; Ilharco et al.,

2022). Other methods include designing the specific neural architecture (Huang et al., 2020; Qiu and Cohen, 2022; Cao et al., 2018), summary ranking (Falke et al., 2019; Liu et al., 2022) and *post-hoc* correction (Zhu et al., 2021; Dong et al., 2020; Cao et al., 2020; Zhao et al., 2020). However, there is still a limited understanding of the effectiveness of these methods in cross-lingual transfer.

## 10 Conclusion

We investigate how to measure and mitigate hallucinations of summarisation models in cross-lingual transfer scenarios. We first propose a multilingual metric, mFACT, to facilitate the evaluation of faithfulness in low-resource languages. By virtue of this new metric, we find empirical evidence that while common cross-lingual transfer methods benefit summarisation performance, they amplify hallucinations compared to monolingual counterparts. We also point out that faithfulness in summarisation for languages other than English is still challenging for multilingual large language models. Finally, with the aim of reducing these hallucinations, we adapt several monolingual methods to cross-lingual transfer and propose a new method based on weighting the loss according to the mFACT score of each training example. Based on both automated metrics and human evaluation, we demonstrate that mFACT is the most reliable metric in detecting hallucinations in multiple languages. Moreover, compared to a series of state-of-the-art baselines, we find that summaries produced by loss weighting achieve higher performance and abstractiveness, competitive faithfulness, and a higher alignment with human preferences. We hope that this work will attract more attention from the community to the phenomenon of hallucination in languages different from English and facilitate future research by establishing evaluation metrics and baselines.

## Limitations

We use machine translation to construct the faithfulness classification dataset for training the faithfulness metrics in target languages. The required resources may constrain the feasibility of extending mFACT to other languages. The quality of the learned metrics may also be limited by the propagation of errors during translation, especially for languages with poor translation performance. Additionally, although the weighted-loss approach is effective in a diverse sample of languages, we note that its gains in faithfulness are not consistent for all languages, as we discussed in Section 8. Finding a method that is equally effective in reducing hallucinations across all languages is still an open research question for future work.

## Ethical Consideration

All human workers participating in our evaluation are informed of the intended use of the provided assessments of summary quality and comply with the terms and conditions of the experiment, as specified by Prolific. In regards to payment, workers from different regions are paid on the same high scale with a wage of £13.5 hourly. This work (and specifically, the human evaluation) has also passed an ethical review by the ethical panel in our institute.

## Acknowledgements

We would like to thank Zheng Zhao and the anonymous reviewers for their helpful feedback. We are grateful for an Apple AI/ML scholarship awarded to Yifu Qiu. We appreciate the use of computing resources through the Baskerville cluster at the University of Birmingham.

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

# A  Appendix

## A.1  Dataset Statistics

We show the dataset statistics for all six used subsets of XL-Sum in table 7.

## A.2  Implementation Details.

**mFACT Classifiers** We implement mFACT with the `transformers` package (Wolf et al., 2020). We train the multilingual BERT model for two epochs, with a batch size of 32 and a learning rate of 5e-5. We set the max input length to 512 and apply truncation to the input article if necessary. The same hyper-parameter settings are applied to all the languages we test.

**Weighted Loss Summarisation Models** We implement our weighted loss model for cross-lingual transfer with `adapter-transformers` package (Pfeiffer et al., 2020a). We use the officially released mBART-50 checkpoint as the base model for equipping language and task adapters.

To train the language adapters, we follow the same adapter architecture and training settings in (Pfeiffer et al., 2020b). We use the batch size of 64, and a learning rate of 1e-4. We train each adapter with 48K update steps.

**Task Adapters** To train the task adapters for summarisation, we set the batch size to 32, the learning rate to 1e-4, label smoothing factor to 0.1. We use the polynomial scheduler for adjusting the learning rate during training, with weighted decay at 0.01 and maximum gradient norm at 0.1. The model is trained for ten epochs, and we set the first 500 update steps as the warm-up stage. We select the best checkpoint following either the best validation ROUGE or the best mFACT score, respectively. During the decoding step for zero-shot cross-lingual transfer, we follow most settings of (Hasan et al., 2021). We apply the beam search with a size of 6, and the minimum/maximum decoding steps are set to 30/84, respectively. The length penalty is applied at 0.6, and we block all repeated tri-grams.

## A.3  Sanity Check for English Faithfulness Metrics

We perform a sanity check experiment and report the results in Table 9 to verify the reliability of these model-based hallucination metrics. We randomly shuffle the alignments of document-summary pairs predicted by the mBART model and the reference. We then feed these misaligned document-summary

pairs into the evaluation models and test their performance. We observe that all hallucination metrics drop considerably, showing that these metrics are indeed sensitive to random summaries and reliable to some extent.

## A.4  Translation Quality Check

Our first experiment is to confirm the effectiveness of mFACT in capturing hallucinations in target languages. To support our method, we conduct a quality check for translation outputs, a comparison of different metrics on our translated faithfulness classification dataset, and an external evaluation of downstream tasks.

Machine translation (MT)-based transfer can arguably suffer from error propagation, where MT tools introduce hallucinations into their outputs. This issue is even more serious in our setting where translating faithful samples is necessary to create the mFACT metric as training with false positives might significantly degrade its quality. To ensure the feasibility of our pipeline to develop mFACT, we first check the translation quality manually. We randomly pick 100 samples from the Chinese positive set and label their faithfulness. Through this sanity check, we found 13 hallucinated samples; however, only 4 of them are caused by poor translation, while the other 9 are due to an incorrect ranking based on the four English metrics. This shows that MT-based transfer is mostly reliable: only a small amount of noise is introduced by MT.

## A.5  Extended Results for Faithfulness Classification

To gain a deeper comprehension of the averaged faithfulness classification results presented in add a reference to Table 1, we analyse the individual language-specific outcomes (Table 10). Across the six language experiments, we consistently observe a significant performance gap between the models trained on the NLI task and those trained on the faithfulness classification task.

The following is the guide for annotators to indicate whether a summary is informative and faithful.

## A.6  Full-model transfer vs. MAD-X transfer

We conduct a comparative study on the performance of summarisation and faithfulness in two cross-lingual transfer approaches: MAD-X style and full-model transfer.

For both MAD-X style and full-model cross-lingual transfer, we observe that cross-lingual trans-

| Language | Family | Doc. Len. | Sum. Len. | Comp% | Prop. A | Prop. C |
|---|---|---|---|---|---|---|
| Chinese | Sino-Tibetan | 859 | 48 | 9.63 | 93.49 | 29.56 |
| French | Indo-European | 743 | 43 | 12.07 | 99.20 | 26.72 |
| Spanish | Indo-European | 1242 | 42 | 11.12 | 84.71 | 42.93 |
| Vietnamese | Austro-Asiatic | 1647 | 50 | 6.09 | / | / |
| Turkish | Turkic | 747 | 44 | 9.33 | / | / |
| Hindi | Indo-European | 1200 | 49 | 8.59 | 90.91 | 31.42 |

Table 7: Dataset statistics for six annotated datasets from XL-Sum (Hasan et al., 2021). We report the document length (*Doc. Len.*), summary length (*Sum. Len.*) and the corresponding compression percentage (*Comp%*). All lengths are measured in the unit of tokens. We also include quality measurements from the XL-Sum paper according to human annotators. *Prop. A* reports the percentage of the summaries that convey the main idea of the document. *Prop. C* reports the percentage of summaries that contain some additional information not presented in the source.

| Document & Translation | Summaries & Translations | mFACT |
|---|---|---|
| 在英法海底隧道工作的19名工人一氧化碳中毒，其中一人情况严重。事故发生时，有大约60名工人正在法国加莱与英国福克斯通之间的海底隧道里更换铁轨。星期天凌晨一名焊工患病，后来被确诊是一氧化碳中毒，另有18名工人也不同程度地中毒。他们被送往当地一家法国医院治疗，其余41名工人都已回家休息。[...] | **MAD-X:** 英法海底隧道内的一名工人中毒,其中一人情况严重,其余41名工人已被送往法国医院治疗。 | 0.12 |
|  | **Weighted Loss:** 事故发生在法国加莱附近海底隧道里,其中一名工人中毒,另有18名工人受伤,其中一人情况严重。 | 0.78 |
| Nineteen workers working in the Channel Tunnel have been poisoned with carbon monoxide, one in serious condition. About 60 workers were replacing rails in the undersea tunnel between Calais, France, [...] Sunday morning, a welder fell ill and was later diagnosed with carbon monoxide poisoning, and 18 other workers were also poisoned to varying degrees. They were sent to a local French hospital for treatment, and the remaining 41 workers have gone home to rest. [...] | **MAD-X:** A worker in the British-French Channel Tunnel was poisoned, one of them in a serious condition, and the remaining 41 workers have been sent to French hospitals for treatment. | 0.12 |
|  | **Weighted Loss:** The accident occurred in the undersea tunnel near Calais, France. One worker was poisoned and 18 workers were injured, one of them in serious condition. | 0.78 |

Table 8: Examples of hallucinations (highlighted in a orange colour) generated by MAD-X, a method for zero-shot cross-lingual transfer, on top of an mBART-50 backbone. In this work, we present 1) the mFACT metric to evaluate the faithfulness of summarisation models in target languages other than English and 2) a loss weighting method to reduce hallucinations in cross-lingual transfer. We highlight the faithful pieces of information produced by our loss weighting but missed by MAD-X in a green colour.

|  | DAE (↑) | QAFE (↑) | ENFS% (↓) | EntFA (↑) |
|---|---|---|---|---|
| Reference | 19.37 | 34.14 | 64.49 | 72.50 |
| Shuffled | 0.00 | 0.15 | 97.57 | 34.63 |
| mBART | 38.40 | 37.82 | 53.24 | 82.67 |
| Shuffled | 8e-5 | 0.13 | 97.42 | 36.12 |

Table 9: Hallucination scores of the reference and mBART's outputs and their corresponding shuffled document-summary pairs (Shuffled). All results are evaluated on the XSum test set. ↑ and ↓ indicate higher and lower hallucination score is better, respectively. QAFE stands for QAFactEval.

fer leads to improved ROUGE scores, but it also results in reduced faithfulness scores. Interestingly, when comparing the effects of Full-model and MAD-X transfer in Figure 3, we see that Full-model transfer exhibit a greater improvement in ROUGE scores. However, this improvement come at the expense of introducing more hallucinations.

In Figure 4, we further compare the performance of Full-model and MAD-X transfer in both zero-

shot and few-shot transfer scenarios. While Full-model transfer demonstrates an advantage in the few-shot transfer scenario compare to MAD-X, MAD-X performs better in the zero-shot scenario. Additionally, regardless of the transfer scenario, MAD-X exhibits a lower occurrence of hallucinations.

## A.7 Validation Faithfulness Curve for Weighted Loss Method

We provide the training curve of MAD-X and our weighted loss method in Figure 5.

Upon examining the curve, it becomes evident that the model trained with the weighted loss consistently exhibits higher faithfulness compared to the MAD-X baseline throughout the entire training process. This observation serves as evidence for the effectiveness of our approach in guiding the model's optimisation towards increased faithfulness by diverting its training away from hallucinated samples.

| | Models | Acc. | Prec. | Recall | F1 |
|---|---|---|---|---|---|
| **Spanish** | XNLI | 54.00 | 76.47 | 10.48 | 18.44 |
| | XNLI-mFACT | 96.40 | 97.13 | 95.56 | 96.34 |
| | mFACT-Transfer | 70.60 | 98.10 | 41.53 | 58.36 |
| | mFACT | 97.00 | 97.55 | 96.37 | 96.96 |
| **Hindi** | XNLI | 53.00 | 74.07 | 8.06 | 14.55 |
| | XNLI-mFACT | 94.20 | 94.33 | 93.95 | 94.14 |
| | mFACT-Transfer | 54.00 | 100 | 7.26 | 13.53 |
| | mFACT | 94.00 | 95.04 | 92.74 | 93.88 |
| **Turkish** | XNLI | 52.40 | 69.23 | 7.26 | 13.14 |
| | XNLI-mFACT | 96.60 | 97.53 | 95.56 | 96.54 |
| | mFACT-Transfer | 59.60 | 100 | 18.55 | 31.29 |
| | mFACT | 97.00 | 97.17 | 96.77 | 96.97 |
| **Vietnamese** | XNLI | 53.40 | 72.73 | 9.68 | 17.08 |
| | XNLI-mFACT | 96.00 | 96.72 | 95.16 | 95.93 |
| | mFACT-Transfer | 68.00 | 97.83 | 36.29 | 52.94 |
| | mFACT | 95.40 | 94.47 | 96.37 | 95.41 |
| **French** | XNLI | 52.60 | 67.74 | 8.47 | 15.05 |
| | XNLI-mFACT | 96.20 | 96.73 | 95.56 | 96.15 |
| | mFACT-Transfer | 65.80 | 98.73 | 31.45 | 47.71 |
| | mFACT | 95.60 | 95.20 | 95.97 | 95.58 |
| **Chinese** | XNLI | 52.20 | 69.57 | 6.45 | 11.80 |
| | XNLI-mFACT | 90.80 | 91.39 | 89.91 | 90.65 |
| | mFACT-Transfer | 67.40 | 98.85 | 34.67 | 51.34 |
| | mFACT | 92.60 | 92.71 | 92.34 | 92.53 |

Table 10: Classification performance on our translated faithfulness dataset for all target languages.

## A.8 Prompts Used for Multilingual LLM's Summarisation

We show the prompt templates used for all languages in our LLM's summarisation experiments in Figure 6.

## A.9 Assembling Metrics for mFACT does better than Single Metric

We conducted an additional experiment to support our assembling design of mFACT. Rather than averaging four metrics, we individually apply single English metric - DAE, QAFE, ENFS, and EntFA — to rank the XSum dataset and train a multilingual classifier similar to mFACT-Transfer without translation, denoted as DAE-T, QAFE-T, ENFS-T, and EntFA-T.

To examine mFACT with other metrics originating from each single metric, we extend the human evaluation results in Table 6. We compare these four metrics with mFACT-Transfer, and again we measure the Pearson and Spearman correlations to human annotations.

In Table 11, we find mFACT consistently

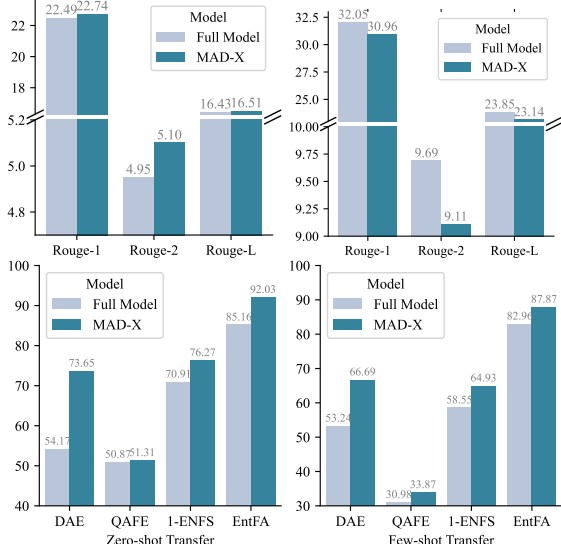

Figure 4: Comparison of Full-model and MAD-X cross-lingual transfer in ROUGE and faithfulness. The left column is the zero-shot performance, and the right column is the few-shot performance. We provide the average scores over all six languages.

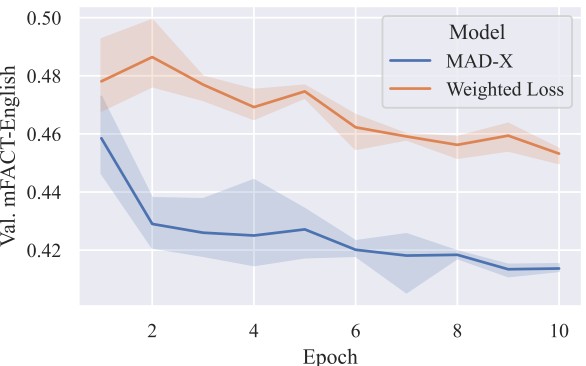

Figure 5: Validation mFACT scores curve for each model's training dynamics. Weighted loss consistently outperforms MAD-X in terms of faithfulness during the whole training period.

emerges with the highest human correlation when compared to other four metrics. This observation underscores mFACT's better correlation with human evaluations. The reason could be relying on a single metric can introduce biased preference in models and a lack of diversity for captured hallucinations. In general, multiple teacher models lead to a robust, unbiased process (Wu et al., 2021; Ilichev et al., 2021). Using diverse metrics in mFACT's training helps the classifier detect various hallucination types - our inverse transfer experiments (Table 2) also show mFACT's promising correlations with both intrinsic and extrinsic hallucination metrics.

| Language | Prompt |
|----------|--------|
| English | Summarize the given article: ${en_article} |
| Chinese | 总结给定的文章: ${zh_article} |
| French | Résumer l'article donné: ${fr_article} |
| Spanish | Resume el artículo dado: ${es_article} |
| Hindi | दिए गए लेख को संक्षेप में प्रस्तुत करें : ${hi_article} |

Figure 6: Prompt templates we used for conducting summarisation experiments with multilingual large language models.

| | Pearson | | Spearman | |
|---|---|---|---|---|
| | $\rho$ | $p$ **value** | $\rho$ | $p$ **value** |
| mF-T | **0.44**$^*$ | 0.00 | **0.36**$^*$ | 0.01 |
| DAE-T | 0.34$^*$ | 0.01 | 0.33$^*$ | 0.01 |
| QAFE-T | 0.25 | 0.07 | 0.29$^*$ | 0.04 |
| ENFS-T | 0.24 | 0.07 | 0.29$^*$ | 0.04 |
| EntFA-T | 0.35$^*$ | 0.01 | **0.36**$^*$ | 0.01 |

Table 11: Correlation with human preferences for mFACT and four transferred metrics developing from single metric. We again calculate both Pearson and Spearman statistics on document–summary pairs from all six languages to ensure that the sample size is significant.

## A.10 Strategy for Selecting Best Model Checkpoint

Table 12 compares the summarisation model performance when we select the model checkpoint with the best ROUGE-1 or the best mFACT score. We find that under both strategies, the weighted loss model can achieve better ROUGE and faithfulness scores in most languages. However, similar to other works (Choubey et al., 2021; Aharoni et al., 2022), selecting the model checkpoint with the best validation faithfulness score has a higher positive contribution to model's faithfulness.

## A.11 Distributions of Faithfulness and Abstractiveness for All Languages

We show the distributions for the percentage of novel 2-grams and mFACT scores for all six languages in Figure 7.

| L. | Method | Best Validation ROUGE-1 | | | | | | | Best Validation mFACT-English | | | | | | |
|---|---|---|---|---|---|---|---|---|---|---|---|---|---|---|---|
| | | R-1 | R-2 | R-L | mF | mF-T | bi% | tr% | R-1 | R-2 | R-L | mF | mF-T | bi% | tr% |
| Chinese | MAD-X | 27.97 | 14.10 | 19.95 | 38.29 | 33.99 | 28.23 | 44.15 | 29.59 | 14.86 | 20.61 | 39.62 | 35.08 | 21.62 | 34.37 |
| | CAPE | 28.15 | 14.57 | 20.74 | **42.56** | **39.59** | 27.12 | 42.11 | 29.64 | 14.80 | 20.58 | 38.83 | 34.01 | 20.11 | 32.32 |
| | TVN | 19.64 | 9.64 | 14.27 | 36.05 | 31.28 | 9.98 | 16.68 | 29.68 | 14.75 | 20.32 | 38.53 | 32.61 | 17.67 | 28.76 |
| | Dexpert | 27.98 | 14.10 | 19.96 | 38.17 | 33.97 | 28.21 | 44.12 | 29.59 | 14.86 | 20.61 | 39.63 | 35.08 | 21.62 | 34.37 |
| | Ours | **30.81** | **16.14** | **22.04** | 42.50 | 35.88 | **36.38** | **54.85** | **31.24** | **16.13** | **22.06** | **43.16** | **37.85** | **30.16** | **47.02** |
| Spanish | MAD-X | 20.77 | 4.89 | 15.06 | 20.57 | 29.25 | 33.91 | 50.47 | 23.36 | 5.13 | 16.34 | 21.87 | 29.36 | 21.98 | 34.20 |
| | CAPE | 21.16 | 4.91 | 15.28 | 22.03 | 31.02 | 29.99 | 45.22 | 23.24 | 5.01 | 16.24 | 21.65 | 29.40 | 19.98 | 31.29 |
| | TVN | 20.27 | 4.94 | 14.98 | **28.61** | **36.05** | 18.92 | 30.66 | 23.53 | 5.06 | 16.48 | 23.82 | 30.54 | 18.02 | 28.60 |
| | Dexpert | 20.75 | 4.88 | 15.04 | 20.46 | 29.25 | 33.85 | 50.42 | 23.36 | 5.13 | 16.34 | 21.88 | 29.36 | 21.98 | 34.20 |
| | Ours | **22.62** | **5.54** | **16.31** | 22.09 | 32.50 | **41.90** | **60.49** | **24.30** | **6.10** | **17.41** | 23.83 | 33.31 | **34.54** | **51.69** |
| Hindi | MAD-X | 20.08 | 5.44 | 15.10 | 22.75 | 16.42 | 34.96 | 51.35 | 25.51 | 7.78 | 19.07 | 28.41 | 19.32 | 25.57 | 39.02 |
| | CAPE | 20.97 | **6.11** | **16.07** | 23.50 | 16.45 | 30.95 | 45.81 | **25.80** | **7.85** | **19.20** | 29.11 | 19.53 | 23.77 | 36.58 |
| | TVN | 19.44 | 5.78 | 15.48 | **34.36** | **27.22** | 24.12 | 35.71 | 25.28 | 7.73 | 19.15 | **32.76** | **24.61** | 21.57 | 33.28 |
| | Dexpert | 19.96 | 5.74 | 15.32 | 22.79 | 16.43 | 34.97 | 51.37 | 25.51 | 7.78 | 19.07 | 28.40 | 19.32 | 25.57 | 39.02 |
| | Ours | **20.99** | 5.73 | 15.70 | 20.32 | 13.75 | **47.44** | **65.9** | 24.47 | 7.46 | 18.48 | 28.48 | 19.52 | **34.86** | 50.99 |
| Turkish | MAD-X | 17.16 | 5.56 | 14.00 | 30.16 | 21.08 | 43.59 | 60.56 | **17.22** | **6.33** | **14.59** | 33.24 | 25.53 | 38.72 | 54.12 |
| | CAPE | **17.66** | 5.72 | 14.33 | 29.88 | 22.50 | 39.32 | 56.45 | 17.12 | 6.23 | 14.55 | **35.04** | **26.69** | 36.47 | 51.47 |
| | TVN | 17.49 | 5.82 | **14.63** | **34.05** | **22.91** | 33.68 | 48.82 | 16.95 | 6.28 | 14.49 | 34.56 | 25.97 | 34.05 | 48.91 |
| | Dexpert | 17.17 | 5.56 | 14.01 | 28.47 | 21.10 | 43.64 | 60.61 | **17.22** | **6.33** | **14.59** | 33.22 | 25.53 | 38.72 | 54.12 |
| | Ours | **17.66** | **5.95** | 14.44 | 29.80 | 22.21 | **54.91** | **72.2** | 17.16 | 6.28 | 14.46 | 34.91 | 25.83 | **45.34** | 61.50 |
| Vietnamese | MAD-X | 25.85 | 11.85 | 19.45 | 35.09 | 34.10 | 33.99 | 50.73 | 27.23 | 12.57 | 20.32 | 36.64 | 37.75 | 27.40 | 42.67 |
| | CAPE | 26.20 | 11.96 | 19.65 | 36.98 | 36.79 | 30.68 | 46.77 | 27.01 | 12.45 | 20.15 | 36.71 | 37.79 | 25.89 | 40.64 |
| | TVN | 24.40 | 11.32 | 18.83 | 36.94 | 34.85 | 25.69 | 39.38 | 26.73 | 12.36 | 20.07 | **38.41** | **39.34** | 25.86 | 40.48 |
| | Dexpert | 25.86 | 11.85 | 19.46 | 35.15 | 34.19 | 33.99 | 50.74 | 27.23 | 12.57 | 20.32 | 36.61 | 37.75 | 27.40 | 42.67 |
| | Ours | **27.01** | **12.35** | **20.17** | **37.56** | **38.32** | 35.32 | 52.02 | **27.76** | **12.86** | **20.83** | 38.27 | 38.02 | **30.23** | 46.27 |
| French | MAD-X | 25.82 | 8.31 | 19.10 | 38.21 | 40.79 | 27.38 | 42.21 | 26.02 | 7.97 | 19.02 | 38.71 | **42.66** | 18.88 | 30.29 |
| | CAPE | 25.83 | 8.40 | 19.08 | 38.30 | 41.44 | 26.11 | 40.44 | 25.75 | 7.93 | 18.80 | 37.54 | 40.91 | 18.00 | 28.88 |
| | TVN | 24.42 | 7.90 | 18.34 | **42.27** | **44.60** | 16.97 | 28.35 | 25.54 | 7.86 | 18.71 | 38.18 | 41.74 | 17.37 | 27.68 |
| | Dexpert | 25.82 | 8.31 | 19.10 | 38.21 | 40.82 | 27.37 | 42.19 | 26.02 | 7.97 | 19.02 | **38.74** | **42.66** | 18.88 | 30.29 |
| | Ours | **27.63** | **9.21** | **20.28** | 35.57 | 39.47 | **41.52** | **59.22** | **27.70** | **9.09** | **20.27** | 36.83 | 39.75 | **33.81** | 50.57 |

Table 12: Automatic evaluation for **zero-shot** performance in English-to-others cross-lingual transfer direction while selecting the checkpoint with the best validation ROUGE-1 and the best validation mFACT score. We run all model results with three different random seeds. mF stands for mFACT and mF-T stands for mFACT-Transfer. tr% and bi% are the percentage of novel tri-gram and bi-gram, respectively.

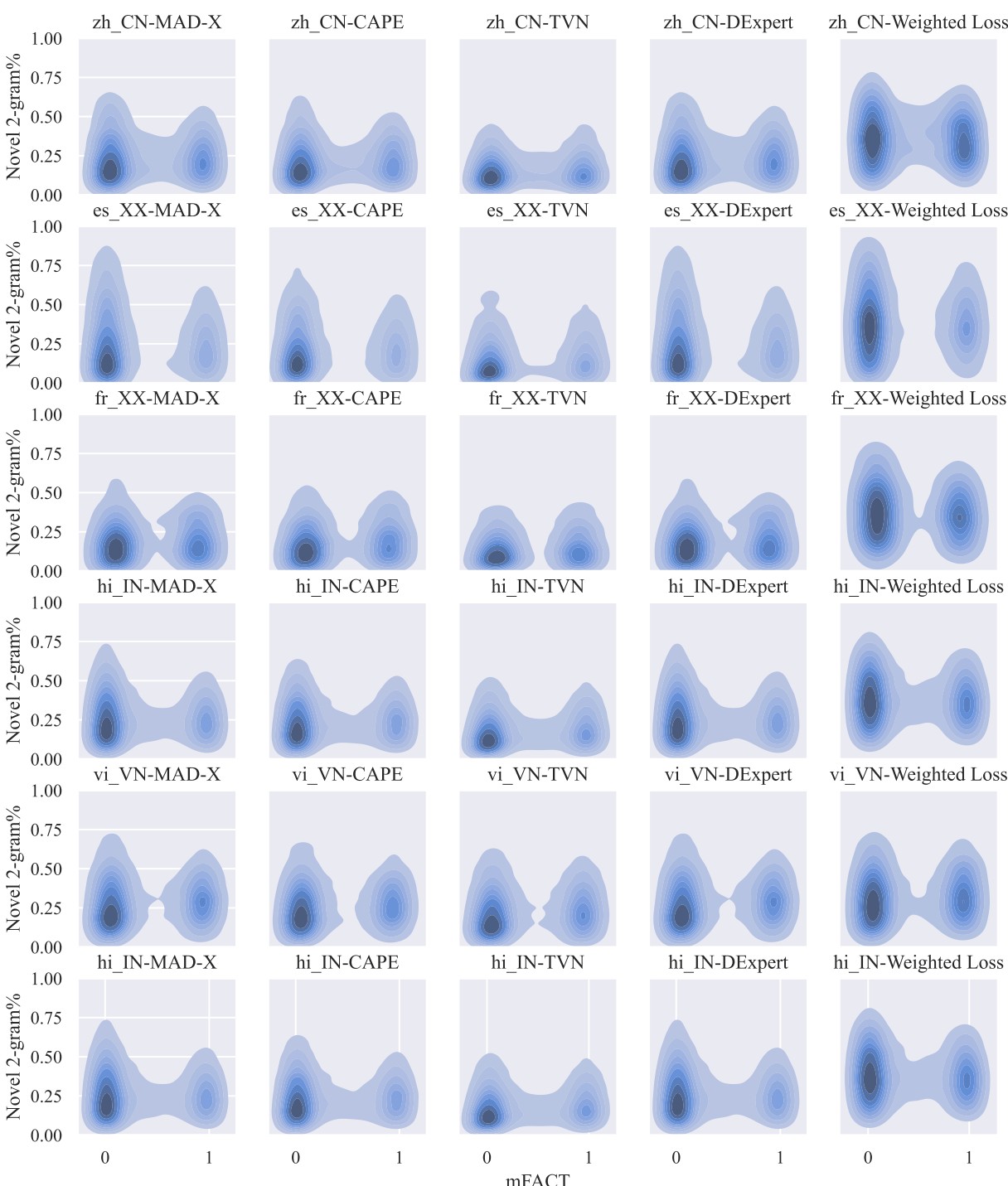

Figure 7: Distributions for Novel 2-gram% and mFACT scores for all hallucination reduction methods in cross-lingual transfer for six languages.