# OpenReview forum: "Detecting and Mitigating Hallucinations in Multilingual Summarisation"
_EMNLP/2023/Conference — EMNLP 2023 Main_

### Official Review · Reviewer_Pf1A · 2023-07-30

**Soundness:** 4

**Excitement:**

4: Strong: This paper deepens the understanding of some phenomenon or lowers the barriers to an existing research direction.

**Paper Topic And Main Contributions:**

The paper investages how to measure and mitigate hallucinations of summarisation models in cross-lingual transfer scenarios. The main contributions are:
1. In order to detect the hallucination in other languages, the peper proposed  mFACT, a multilingual faithful metric developed based on four English faithfulness metrics by firstly rank top-k and bottom-k hallucinated samples and train a classifier based on the selected samples.
2. Before mitigate hallucination, the paper investaged the hallucination in a cross-lingual transfer setting, especially in abstractive summarization. The authors found cross-lingual methods can improve the performance for low-resource summarisation but they also amplify hallucinations.
3. The paper proposed a novel method to migitage hallucination in cross-lingual setting by weighting the loss of training samples based on their faithfulness score. And the results show that the propsoed method can effecitvely reduce hallucination.

**Reasons To Accept:**

1. The paper is the first work that study hallucination in a cross-lingual transfer setting, which can potentially facilitate related research.
2. The proposed mFACT model-based metric can accurately detect hallucination (95.3%)
3. The proposed mitigation approach generally reduce the hallucination in the generated summary.

**Reasons To Reject:**

1. The proposed mFACT metric need lots of data samples in English for training the model, which could be lack of generalization ability.

**Reproducibility:**

4: Could mostly reproduce the results, but there may be some variation because of sample variance or minor variations in their interpretation of the protocol or method.

**Reviewer Confidence:**

4: Quite sure. I tried to check the important points carefully. It's unlikely, though conceivable, that I missed something that should affect my ratings.

---

> ### Author Rebuttal · Authors · 2023-08-28
>
> We thank the reviewer for acknowledging that mFACT can contribute to the NLP research community and their comments.
>
> **Q**: The proposed mFACT metric need lots of data samples in English for training the model, which could be lack of generalisation ability.
>
> **A**: While we use 40K document-summary pairs, they can be automatically labelled by using English metrics, which achieves promising results on both our classification task, inverse-transfer verification and human evaluation. In terms of data-related costs, we are on par with other popular metrics, e.g., the NLI-based methods we use as baselines [1, 2], which require training specialised models to evaluate faithfulness in the monolingual case. Having said that, we agree that mFACT data requirements might be a limitation in some cases, and we will mention it in the "Limitations" section.
>
> References:
>
> [1] On Faithfulness and Factuality in Abstractive Summarization. https://aclanthology.org/2020.acl-main.173.pdf
>
> [2] Evaluating the Factual Consistency of Abstractive Text Summarization. https://arxiv.org/pdf/1910.12840.pdf

---

### Official Review · Reviewer_HygN · 2023-08-03

**Typos Grammar Style And Presentation Improvements:** 1. If you weight the loss according t…
**Soundness:** 3

**Excitement:**

4: Strong: This paper deepens the understanding of some phenomenon or lowers the barriers to an existing research direction.

**Paper Topic And Main Contributions:**

This paper primarily addresses the issue of hallucination in multilingual summarization. The authors introduce a model-based metric to facilitate the evaluation and detection of hallucinations in low-resource languages. Additionally, they believe that crosslingual transfer methods tend to introduce more hallucinations while improving summarization performance. To address this, they propose a weighted loss method that leverages the mFACT metric.

**Questions For The Authors:**

Would you train different classifiers to obtain mFACT for each language, or would you directly train a multilingual classifier with mixed datasets?

**Reasons To Accept:**

1. The authors propose a new multilingual faithful metric mFACT, which can be used in low-resource languages. They have conducted experiments to show the effectiveness of mFACT.
2. They employ the mFACT metric to design a method for mitigating hallucination.
3. They analyze the experiment results in detail.
4. The paper is mostly well-written and easy to follow.

**Reasons To Reject:**

1. I don't think the experiments support the second contribution of the paper. They ignore the development of the LLMs. Previous work shows that the LLMs learn from English resources and improve performance in low-resource languages. The authors haven't conduct experiments to prove that hallucinations emerge more on low-resource languages than English. And they haven't deployed mFact on such experiments.
I highly recommend the authors report the multilingual performance of LLMs like ChatGPT. If LLMs can achieve better performance than cross-lingual transfer baselines and your methods, proposing a method to mitigate hallucination in a cross-lingual setting will be less meaningful. And previous studies have demonstrated that abstracts generated by ChatGPT rarely contain intrinsic hallucinations.
2. According to the description in Section 2.1, your pipeline for creating mFACT is too complex. It is model-based, and it need to be trained on the XSum dataset. The Google API is also needed. So, I think mFACT is hard to be applied. The authors should provide more details about the cost to dispel users' concerns.
3. The weighted loss approach is not easily used to train large models. So, I don't think the mitigating hallucination method will contribute much.

**Reproducibility:**

4: Could mostly reproduce the results, but there may be some variation because of sample variance or minor variations in their interpretation of the protocol or method.

**Reviewer Confidence:**

4: Quite sure. I tried to check the important points carefully. It's unlikely, though conceivable, that I missed something that should affect my ratings.

---

> ### Author Rebuttal · Authors · 2023-08-28
>
> We thank the reviewer for their comments.
>
> **Q**: The authors should compare their results with ChatGPT, as it can achieve better performance than cross-lingual transfer baselines, which will make your contribution less meaningful.
>
> **A**: We firmly assert that employing ChatGPT or any analogous black-box model as a foundational reference in a scientific inquiry comes with substantial drawbacks. For example, recent research has exposed the significant variability in ChatGPT's performance over time, undermining the reproducibility of experiments [1]. Moreover, assessing a black-box model poses the challenge of data contamination, an issue of great magnitude with such data-intensive models [2, 3]. For more reasons to avoid using black-box models as baselines, we kindly refer the reviewer to a more comprehensive blog post written by one of ACL 2023 organizers, Anna Rogers [4].
>
> With that being acknowledged, even a potent black-box LLM like ChatGPT exhibits inferiority when pitted against more compact, task-specific models. A case in point is [5]'s demonstration that ChatGPT lags behind state-of-the-art summarization models for monolingual English summarization. This disparity becomes more pronounced when assessing less commonly spoken languages [6]. As for the faithfulness of such models, recent studies demonstrated that ChatGPT is far from being hallucination-free when using English or low-resource languages as input [7], where it struggles mostly with extrinsic hallucinations [8]. The inverse transfer results we provide in Table 2 show that mFACT correlates well with both intrinsic metrics (DAE and QAFE) and extrinsic metrics (ENFS and EntFA), which are still a significant issue even for strong black-box models such as ChatGPT.
>
> References:
>
> [1] Can we trust the evaluation on ChatGPT? https://arxiv.org/pdf/2307.09009.pdf
>
> [2] Can we trust the evaluation on ChatGPT? https://arxiv.org/pdf/2303.12767.pdf
>
> [3] Speak, Memory: An Archaeology of Books Known to ChatGPT/GPT-4. https://arxiv.org/pdf/2305.00118.pdf
>
> [4] Closed AI Models Make Bad Baselines. https://hackingsemantics.xyz/2023/closed-baselines/
>
> [5] A Systematic Study and Comprehensive Evaluation of ChatGPT on Benchmark Datasets. https://aclanthology.org/2023.findings-acl.29.pdf
>
> [6] ChatGPT Beyond English: Towards a Comprehensive Evaluation of Large Language Models in Multilingual Learning. https://arxiv.org/pdf/2304.05613.pdf
>
> [7] Hallucinations in Large Multilingual Translation Models. https://arxiv.org/pdf/2303.16104.pdf
>
> [8] A Multitask, Multilingual, Multimodal Evaluation of ChatGPT on Reasoning, Hallucination, and Interactivity. https://arxiv.org/pdf/2302.04023.pdf
>
> **Q**: Your pipeline for creating mFACT is too complex. It is model-based, and it need to be trained on the XSum dataset. The Google API is also needed. The authors should provide more details about the cost to dispel users' concerns.
>
> **A**: We want to emphasise that we do *not* use the Google API at inference time but rather use it offline once to obtain the translated training example. We will clarify it better in our paper. As for using the XSum dataset for obtaining our metric, we believe the efforts we invest are on par with other popular metrics, e.g., the NLI-based methods we use as baselines [1, 2], which require training specialized models to evaluate faithfulness. Leveraging translation for developing datasets/resources is a common practice in cross-lingual and multilingual summarization [3], and future research with limited resources can be carried out using an open-source translator if additional adaptation is needed or use our mFACT-transfer for easily extending other languages using zero-shot cross-lingual transfer. For each language, our method has 92M characters to translate (~1000 GBP) on average. We will add these details to our paper.
>
> References:
>
> [1] On Faithfulness and Factuality in Abstractive Summarization. https://aclanthology.org/2020.acl-main.173.pdf
>
> [2] Evaluating the Factual Consistency of Abstractive Text Summarization. https://arxiv.org/pdf/1910.12840.pdf
>
> [3] NCLS: Neural Cross-Lingual Summarization. https://aclanthology.org/D19-1302/
>
> **Q**: The weighted loss approach is not easily used to train large models. So, I don't think the mitigating hallucination method will contribute much.
>
> **A**: As we established in our first answer, we believe that compact, task-specific models are also of great importance, as they achieve state-of-the-art across many tasks. In addition, there is a vast body of research aiming to fine-tune LLMs in a more cost-efficient manner. Two widely used methods for that are LoRA (Low-Rank Adaptation) [1] and its quantized version QLoRA [2]. We believe that such advances will allow researchers to use our method for LLMs as well.
>
> Furthermore, the weighted loss approach shows the potential to reduce hallucinations by acting as a "soft" sample selector during training. This adaptability can be extended to address other text attributes, such as bias and human preference, using an appropriate rating model. The weighted loss technique, characterised by its simplicity and effectiveness, outperforms existing benchmarks. Its ease of use, reproducibility, and lightweight nature position it as a valuable baseline for advancing research in precise multilingual summarization.
>
> References:
>
> [1] LORA: LOW-RANK ADAPTATION OF LARGE LANGUAGE MODELS. https://arxiv.org/pdf/2106.09685.pdf
>
> [2] QLORA: Efficient Finetuning of Quantized LLMs. https://arxiv.org/pdf/2305.14314.pdf
>
> **Q**: Would you train different classifiers to obtain mFACT for each language, or would you directly train a multilingual classifier with mixed datasets?
>
> **A**: We trained different classifiers, as we believe this strategic choice safeguards against the risk of classifier degradation and aligns with our commitment to ensuring the robustness and adaptability of mFACT to varied languages and contexts. The second option shares semantic content across translated instances requires a careful balance to avoid the classifier overfitting to recurring hallucinated errors in multiple languages, which can compromise its ability to generalise effectively across diverse linguistic contexts.

---

### Official Review · Reviewer_1ZfH · 2023-08-05

**Soundness:** 3

**Excitement:**

4: Strong: This paper deepens the understanding of some phenomenon or lowers the barriers to an existing research direction.

**Paper Topic And Main Contributions:**

In multilingual summarization task, the hallucination becomes more severe in cross lingual setting, where the supervised data is in low resource settings. This paper studies this problem, with the goal of reducing the hallucination in cross lingual transfer learning. (from English source language to different target languages).

To address the research problem, authors first propose a model-based metric (mFact) to measure the faithfulness of the document-summary pair. To train the model predicting mFact measure, first, pseudo faithful and pseudo hallucinated document-summary pairs are selected by an averaged empirical metric. Then, the document-summary pairs are translated into target language, and finally a multilingual BERT is fine-tuned based on the data generated. The empirical metric is obtained by averaging 4 model-based metric to rank the document-summary pair from faithful to hallucinated.

With the support of mFact, the training instance in cross lingual summarization is weighted according to the mFact metric, which helps to improve the summarization quality and reduce the hallucination. Experiments are conducted on XLSum. The experiments first compare mFact with XNLI metric (for NLI task) to verify the effectiveness of mFact in summarization task. Then mFact is adopted in the weighted loss, which reports better Rouge in Chinese, Spanish, Vietnamese, French, but a little mixed in Hindi and Turkish.

**Questions For The Authors:**

(1) One thing wish to know is that if "averaging" is the best way to assemble the 4 model-based metrics. What if each individual metric is used to train mFact alone, will mFact beat all the metrics when they are trained alone?

(2) in Table 5, is there result on the human evaluated correlation to compare mFact with the other 4 metrics?

(3) any chance to give more discussion on the result in Table 4? Appreciate if more discussion can be provided on Hindi and Turkish language.

**Reasons To Accept:**

(1) propose mFact, a model-based metric for faithfulness/hallucination measurement

(2) extensive experiments are conducted to support the paper (in the main content and also in appendix)

**Reasons To Reject:**

Not necessarily strong negative points:

(1) for mFact, wish to see more comparison. For example, in addition to XNLI, any chance to include other baselines for comparison?


**Reproducibility:**

3: Could reproduce the results with some difficulty. The settings of parameters are underspecified or subjectively determined; the training/evaluation data are not widely available.

**Reviewer Confidence:**

3: Pretty sure, but there's a chance I missed something. Although I have a good feel for this area in general, I did not carefully check the paper's details, e.g., the math, experimental design, or novelty.

---

> ### Author Rebuttal · Authors · 2023-08-28
>
> We thank the reviewer for acknowledging that mFACT can contribute to the NLP research community and their comments. We hope the additional experiments we conducted will ease the main soundness concerns.
>
> **Q**: For mFact, wish to see more comparison. For example, in addition to XNLI, any chance to include other baselines for comparison?
>
> **A**: There are currently only a handful of established baselines dedicated specifically to multilingual faithfulness evaluation, as evidenced by the scarcity of reported alternatives in the existing literature. We are aware of a recent contribution at ACL 2023 [1], where XNLI is repurposed as an auxiliary task for faithfulness classification due to its linguistic similarity to our task. This underlines the inherent connection between Natural Language Inference (NLI) and faithfulness assessment [2].
>
> However, we firmly believe in delineating the nuances that set NLI apart from faithfulness classification, particularly in the context of hallucination detection. NLI, while pertinent, might not holistically capture the intricate aspects of faithfulness evaluation as we show in this paper. A specialised metric tailored to the challenges of hallucination detection is crucial for robust future research in this domain.
>
> References:
>
> [1] Multilingual Summarization with Factual Consistency Evaluation. Aharoni et al., ACL 2023 Findings.
>
> [2] Evaluating the Factual Consistency of Abstractive Text Summarization. EMNLP 2020.
>
> **Q**: One thing wish to know is that if "averaging" is the best way to assemble the 4 model-based metrics. What if each individual metric is used to train mFact alone, will mFact beat all the metrics when they are trained alone?
>
> **A**: We believe that relying on a single metric can introduce biased preference in models and a lack of diversity for captured hallucinations. In general, multiple teacher models lead to a robust, unbiased process [1, 2]. Using diverse metrics in mFACT's training helps the classifier detect various hallucination types - our inverse transfer experiments (Table 2) show mFACT's promising correlations with *both* intrinsic and extrinsic hallucination metrics.
>
> We conducted an additional experiment to support our argument. Rather than averaging four metrics, we individually apply single English metrics - DAE, QAFE, ENFS, and EntFA — to rank the XSum dataset and train a multilingual classifier similar to mFACT, denoted as DAE-M, QAFE-M, ENFS-M, and EntFA-M. While the faithfulness classification is not important by itself, we provide the results here to first show that a reliable classifier can be trained for all of the metrics as a sanity check.
>
> | Model     | Acc.  | Prec.  | Recall | F1    |
> |-------------|-------|--------|--------|-------|
> | mFACT-Transfer | 96.0 | 96.34  | 95.56  | 95.95 |
> | DAE-M       | 95.2  | 95.16  | 95.16  | 95.16 |
> | QAFE-M      | 87.2  | 83.09  | 93.15  | 87.83 |
> | ENFS-M      | 96.8  | 98.33  | 95.16  | 96.72 |
> | EntFA-M     | 96.8  | 94.27  | 99.6   | 96.86 |
>
> To further verify our intuition that mFACT's aggregation of multiple metrics is primed to capture a wider array of hallucinations, we execute an additional inverse transfer experiment (XL-Sum Chinese -> XSum English) using classifiers trained with signals from single metrics. We chose to focus on the EntFA metric, as its classifier is the most reliable of all the examined metrics (96.86 F1).
> The results are presented below:
>
> |      Model    | DAE↑   | QAFE↑  | ENFS↓  | EntFA↑ |
> |----------|--------|--------|--------|--------|
> | MAD-X    | 83.38  | 69.69  | 16.38  | 95.6   |
> | mFACT-T  | **88.01** | **78.78** | **12.11** | **96.96** |
> | EntFA-M  | 83.47  | 68.57  | 15.98  | 95.52  |
>
>
> From these outcomes, we observe that training the classifier with a single metric leads to inferior results across all the metrics, including the EntFA in which the EntFA-M model is specialised. We also note that even though EntFA-M can capture extrinsic hallucination well, it still has a limited ability to capture intrinsic hallucination (see weak DAE and QAFE improvements), showing the importance of assembling multiple metrics to capture varied types of hallucinations. We finally incorporate these 4 metrics into human evaluation as discussed in the next response.
>
> References:
>
> [1] One Teacher is Enough? Pre-trained Language Model Distillation from Multiple Teachers. Wu et al., ACL Findings 2021.
>
> [2] Multiple Teacher Distillation for Robust and Greener Models. Ilichev et al., RANLP 2021.
>
>
> **Q**: in Table 5, is there result on the human evaluated correlation to compare mFact with the other 4 metrics?
>
> **A**: To examine mFACT with four other metrics in the actual model's outputs rather than a silver-reference test set, we extend the existing results in Table 5. Using the specialised models we built (see details in the previous answer), we measure the Pearson and Spearman correlations to human annotation:
>
> | Metric    | Pearson (rau) | P-val | Spearman (rau) | P-value |
> |-----------|---------------|-------|----------------|---------|
> | mFACT-T   | **0.44**      | 0     | **0.36**       | 0.01    |
> | DAE-M     | 0.34          | 0.01  | 0.33           | 0.01    |
> | QAFE-M    | 0.25          | 0.07  | 0.29           | 0.04    |
> | ENFS-M    | 0.24          | 0.07  | 0.29           | 0.04    |
> | EntFA-M   | 0.35          | 0.01  | **0.36**       | 0.01    |
>
>
> mFACT consistently emerges with the highest human correlation when compared to these four metrics. This observation underscores mFACT's robustness and its capacity to better correlate with human evaluations.
>
> **Q**: "mFact is adopted in the weighted loss, which reports better Rouge in Chinese, Spanish, Vietnamese, French, but a little mixed in Hindi and Turkish." - any chance to give more discussion on the result in Table 4? Appreciate if more discussion can be provided on Hindi and Turkish language.
>
> **A**: Your question is intriguing. In short, we currently lack a definitive explanation for these performance variations in specific languages. However, we can suggest potential reasons for further investigation.
> - Firstly, discrepancies could arise due to varying translation quality from the Google API across languages, impacting performance negatively for languages with lower translation quality.
> - Secondly, the pre-training of our used multilingual models (e.g., mBERT and mBART) might underrepresent certain languages, leading to poorer results in our downstream task.
> - Thirdly, this mixed performance in ROUGE and faithfulness (we see weighting-loss training improves faithfulness) could also be generally attributed to the disparity between the objectives of enhancing ROUGE scores and faithfulness [1,2,3]. And this is because the heuristic-based dataest construction naturally includes hallucinations in even the testing set [4,5]. We'll include these discussions in Section 7 for context.
>
> References:
>
> [1] CaPE: Contrastive Parameter Ensembling for Reducing Hallucination in Abstractive Summarization
>
> [2] Reducing Quantity Hallucinations in Abstractive Summarization
>
> [3] Improving Factuality of Abstractive Summarization without Sacrificing Summary Quality
>
> [4] Benchmarking Large Language Models for News Summarization
>
> [5] Understanding Factuality in Abstractive Summarization with FRANK: A Benchmark for Factuality Metrics

---

### Meta-Review · Area_Chair_655a · 2023-09-19

**Recommendation:** 4

**Metareview:**

The proposed approach focuses on addressing hallucination issues in multilingual summarization, particularly in cross-lingual transfer learning with low-resource languages. It introduces a model-based metric called mFact to measure the faithfulness of document-summary pairs, helping to reduce hallucinations. This metric is created through a meticulous process involving selecting pseudo-faithful and hallucinated pairs, translating them into the target language, and fine-tuning a multilingual BERT model. By leveraging mFact, the approach introduces a weighted loss method to improve summarization quality and reduce hallucinations.

Pro:
- Introduction of the novel mFact metric, which is quite novel for studying hallucination in cross-lingual settings
- Extensive experiments validate its effectiveness in enhancing ROUGE scores across multiple languages

Cons:
- There are some concerns regarding the complexity of the mFact metric creation, its generalizability, and the robustness of the hallucination mitigation approach
- Need more insights into the practical application of the mFact metric, including discussions on associated costs and potential challenges, would be valuable.
- Mixed results observed in Hindi and Turkish with no further clarification

Given these points, I would recommend for acceptance of the findings, but not against it being accepted to the main conference if SAC finds it suitable.

---

### Decision · Program_Chairs · 2023-10-07

**Decision:**

Accept-Main

**Comment:**

The proposed approach focuses on addressing hallucination issues in multilingual summarization, particularly in cross-lingual transfer learning with low-resource languages. It introduces a model-based metric called mFact to measure the faithfulness of document-summary pairs, helping to reduce hallucinations. This metric is created through a meticulous process involving selecting pseudo-faithful and hallucinated pairs, translating them into the target language, and fine-tuning a multilingual BERT model. By leveraging mFact, the approach introduces a weighted loss method to improve summarization quality and reduce hallucinations.

Pro:
- Introduction of the novel mFact metric, which is quite novel for studying hallucination in cross-lingual settings
- Extensive experiments validate its effectiveness in enhancing ROUGE scores across multiple languages

Cons:
- There are some concerns regarding the complexity of the mFact metric creation, its generalizability, and the robustness of the hallucination mitigation approach
- Need more insights into the practical application of the mFact metric, including discussions on associated costs and potential challenges, would be valuable.
- Mixed results observed in Hindi and Turkish with no further clarification

Given these points, I would recommend for acceptance of the findings, but not against it being accepted to the main conference if SAC finds it suitable.